# Presence and Diversity of Different Enteric Viruses in Wild Norway Rats (*Rattus norvegicus*)

**DOI:** 10.3390/v13060992

**Published:** 2021-05-26

**Authors:** Sandra Niendorf, Dominik Harms, Katja F. Hellendahl, Elisa Heuser, Sindy Böttcher, Sonja Jacobsen, C.-Thomas Bock, Rainer G. Ulrich

**Affiliations:** 1Robert Koch Institute, Division of Viral Gastroenteritis and Hepatitis Pathogens and Enteroviruses, Department of Infectious Diseases, 13353 Berlin, Germany; HarmsD@rki.de (D.H.); katja.hellendahl@gmx.de (K.F.H.); BoettcherS@rki.de (S.B.); JacobsenS@rki.de (S.J.); BockC@rki.de (C.-T.B.); 2Friedrich-Loeffler-Institute, Federal Research Institute for Animal Health, Institute of Novel and Emerging Infectious Diseases, 17493 Greifswald-Insel Riems, Germany; elisa.heuser@outlook.de (E.H.); rainer.ulrich@fli.de (R.G.U.); 3German Center for Infection Research (DZIF), Partner Site Hamburg-Lübeck-Borstel-Riems, 17493 Greifswald-Insel Riems, Germany

**Keywords:** astrovirus, enterovirus, hepatitis E virus, norovirus, Norway rat, rodent, rotavirus, zoonosis

## Abstract

Rodents are common reservoirs for numerous zoonotic pathogens, but knowledge about diversity of pathogens in rodents is still limited. Here, we investigated the occurrence and genetic diversity of enteric viruses in 51 Norway rats collected in three different countries in Europe. RNA of at least one virus was detected in the intestine of 49 of 51 animals. Astrovirus RNA was detected in 46 animals, mostly of rat astroviruses. Human astrovirus (HAstV-8) RNA was detected in one, rotavirus group A (RVA) RNA was identified in eleven animals. One RVA RNA could be typed as rat G3 type. Rat hepatitis E virus (HEV) RNA was detected in five animals. Two entire genome sequences of ratHEV were determined. Human norovirus RNA was detected in four animals with the genotypes GI.P4-GI.4, GII.P33-GII.1, and GII.P21. In one animal, a replication competent coxsackievirus A20 strain was detected. Additionally, RNA of an enterovirus species A strain was detected in the same animal, albeit in a different tissue. The results show a high detection rate and diversity of enteric viruses in Norway rats in Europe and indicate their significance as vectors for zoonotic transmission of enteric viruses. The detailed role of Norway rats and transmission pathways of enteric viruses needs to be investigated in further studies.

## 1. Introduction

Zoonotic infections are transmitted between vertebrates and humans and vice versa, and can be caused by bacteria, viruses, parasites, or prions. Zoonotic infections cause more than one billion human illnesses and millions of deaths per year, worldwide [1]. Rodents represent the most divergent and species-rich order of mammals [2]. They are known as reservoirs for many different pathogens and are involved in the emergence and dissemination of viruses, bacteria, and protozoa [3]. The rodent-to-human transmission of zoonotic pathogens occurs via direct routes such as biting or via consumption of water or food contaminated by feces or urine of infected animals. Alternatively, the transmission can be indirectly, via vectors such as ticks, fleas, and mites [4].

The Norway or brown rat (*Rattus norvegicus*) was distributed worldwide through human settlement, probably starting from northern China and Mongolia [5]. Norway rats are considered as food pests and as carriers and transmitters of many pathogens. Rats as neozoa also affect the life of different other animals and plants, especially on isles [6]. Norway rats are represented by wildlife animals, but also by laboratory and pet animals. Pathogen screening of Norway rats by specific assays, such as PCR/RT-PCR, isolation, and serological assays resulted in the detection of various pathogens, including zoonotic pathogens, such as *Leptospira interrogans*, *Streptobacillus moniliformis*, cowpox virus, and Seoul orthohantavirus [7,8,9,10]. In addition, Norway rats were found to harbor rat-specific viral agents, such as herpes-, polyoma-, and papillomaviruses that are most likely non-zoonotic [11,12,13]. Furthermore, Norway rats were found to be infested by human pathogenic agents, such as extended spectrum beta-lactamase (ESBL) producing enterobacteria or noroviruses, although rats may not play a role in the propagation of these agents [14,15]. Finally, rat hepatitis E virus (ratHEV) was discovered in Norway rats and other rat species but was only recently found to cause zoonotic transmission and disease in humans [16,17]. Recent high-throughput sequencing resulted in the identification of a large diversity of infectious agents in rat intestine/fecal samples, suggesting multiple co-infections [18,19]. 

Astroviruses belong to the family *Astroviridae,* which includes highly diverse viruses within two genera, *Mamastrovirus* and *Avastrovirus*, and infect mammals and birds, respectively [20]. Astroviruses are single stranded positive sense RNA viruses. The genome is 6.2 to 7.8 kilobases (kb) in size and contains three open reading frames (ORF). ORF1a and ORF1b encode for the non-structural proteins and ORF2 encodes for the capsid protein. In humans, astroviruses cause acute gastroenteritis mainly in children under five years [21] with symptoms that are milder compared to infections with noro- or rotaviruses [22]. In immunocompromised patients, astrovirus infection can cause extra-intestinal disease with severe complications [23,24,25,26,27]. Astrovirus infections have been reported in a broad range of different mammalian and avian hosts [28]. Despite this broad host range, zoonotic infections are rarely reported. Neves and colleagues identified bat-associated astroviruses in three different rodent and shrew species [29]. A canine astrovirus infection in a child with acute gastroenteritis has been described in a study conducted with samples from children with acute gastroenteritis in Nigeria [30].

Noroviruses and sapoviruses belong to the family *Caliciviridae*. They are non-enveloped RNA viruses with a single stranded positive sense RNA genome. The genome of sapo- and noroviruses contains three ORFs. ORF1 encodes for six non-structural proteins including the RNA-dependent RNA-polymerase (RdRp), ORF2 for the capsid protein (VP1) and ORF3 for the minor capsid protein (VP2) [31]. Sapoviruses have been described in many different hosts, such as humans, chimpanzee, pigs, carnivores (sea lions, dogs, minks), bats, and Norway rats [32,33]. Infections with human sapovirus lead to acute gastroenteritis in all age groups and were detected in outbreaks, as well as in sporadic cases [34,35]. Human sapoviruses are divided into four genogroups with 17 genotypes. To date, there is a gap of knowledge about the zoonotic potential of sapoviruses. So far, animal sapoviruses have not been found in humans. Noroviruses have been identified in a wide range of different mammalian and avian hosts [36,37]. Human noroviruses cause up to 1.1 million hospitalizations and over 218,000 deaths per year in children in resource-poor countries [38]. According to a recently published proposal for norovirus classification, they are classified into ten genogroups with 48 genotypes, with human noroviruses belonging to genogroups GI, GII, GIV, GVII, and GIX [39]. So far, human noroviruses have been detected in Norway rats [15,37], dogs [40,41], pigs [42], rhesus monkeys [43], and in wild birds like gulls and crows [37]. To date, no infection of humans with an animal norovirus has been reported, but in some seroprevalence studies, antibodies against bovine and canine noroviruses have been identified [44,45,46]. These data suggest a possible infection with canine- or bovine-specific noroviruses in humans.

Rotaviruses are segmented double stranded RNA viruses that belong to the family *Reoviridae*. The eleven RNA segments encode for six structural viral proteins (VP1-VP4, VP6 and VP7) and for six non-structural proteins (NSP1-NSP6). The mature viral particle consists out of three protein layers, the inner (VP2), the middle (VP6), and the outer layer (VP4 and VP7) [47]. Rotaviruses are classified into ten different groups (RVA-RVJ) based on the genetic variability of the viral protein 6 (VP6) [48]. Rotavirus group A (RVA) strains were further characterized by the sequence variability of the outer layer proteins, i.e., the VP7 glycoprotein (G type) and the protease-sensitive VP4 (P type), which contain relevant epitopes recognized by neutralizing antibodies. To date, 39 G and 55 P genotypes, detected in humans and different animal hosts, have been described by the rotavirus classification working group of the International Committee on Taxonomy of Viruses (ICTV) [49]. RVA is the most important etiological agent of severe diarrhea in children under five years of age, with an estimated 215,000 deaths recorded in 2013 worldwide [50,51]. In the recent years, there has been a growing number of reports describing the interspecies transmission of rotaviruses from animal to humans or among animals [18,52,53,54,55,56]. Recently, novel rotaviruses have been identified in common shrew (*Sorex araneus*) [57]. 

Hepatitis E virus (HEV), family *Hepeviridae* is the most common cause for acute viral hepatitis worldwide. The virus causes more than 20 million infections with more than 3 million symptomatic cases. In 2015, more than 44,000 patients died as a result of a HEV infection [58]. HEV is endemic in resource-poor countries and appears in sporadic autochthonous outbreaks in industrialized nations [59]. HEV has a single stranded positive sense RNA genome with a size of 6.6 to 7.3 kb, containing three ORFs. ORF1 encodes for the non-structural proteins, ORF2 for the capsid protein, and ORF3 for a small polyfunctional protein [60]. An additional ORF4 of 549 nucleotides (nt) that overlaps ORF1 (+1 frame) has been described for ratHEV [61]. Hepeviruses are classified into two genera *Orthohepevirus* and *Piscihepevirus*. Orthohepeviruses are further divided into four distinct species: species *Orthohepevirus A* is further classified into eight genotypes, HEV-1 to HEV-8 [62]. Human pathogenic hepeviruses belong to genotypes HEV-1, HEV-2, HEV-3, and HEV-4 [63]. Avian hepeviruses belong to species *Orthohepevirus B*, whereas hepeviruses, that infect rats, ferrets and minks were classified as *Orthohepevirus C* and bat-associated hepeviruses as *Orthohepevirus D* (reviewed in [64]). Zoonotic infections of HEV-3 and HEV-4 originated from different animals, including swine, wild boar, and deer [65]). Recently, zoonotic symptomatic ratHEV infections of immunosuppressed and immunocompetent patients have been described [17,66]. 

Hepatitis A virus (HAV) is a non-enveloped, single stranded RNA virus that belongs to the genus *Hepatovirus*, family *Picornaviridae* with a genome of approximately 7.5 kb in size, which contains a single polyprotein-coding ORF that is flanked by 5′ and 3′ untranslated regions, as well as a 3′ poly(A) tail [67]. Rodent hepatoviruses have recently been identified in different cricetid and murid rodent species [68].

Enteroviruses belong to the family *Picornaviridae*, and the genus *Enterovirus* is further divided into 15 species, *Enterovirus A* to *Enterovirus L* and *Rhinovirus A* to *Rhinovirus C* [69]. Enterovirus infections lead to a broad range of clinical manifestations, ranging from mild gastrointestinal or respiratory symptoms to severe neurological disorders, like meningitis and acute flaccid paralysis, and in some cases even to death [70]. Some studies reported the presence of human enteroviruses in animal species. A high number of Coxsackie B3 virus RNA (CVB3) was detected in the heart muscle of a chimpanzee that died of myocarditis in a zoo in Denmark [71]. In another study, enteroviruses EV-A90 and EV-B107 were isolated from fecal samples, collected from wild living chimpanzees in Gabon [72]. These studies showed the spillover of a human enterovirus to animal species. In addition, one study confirmed co-circulation of enteroviruses between apes and humans, indicating the zoonotic potential of enteroviruses [73]. Antibodies against human enterovirus types E11, EV-A76, and EV-D94 were detected in chimpanzees, gorillas, and Old-World monkeys [74]. Furthermore, a recent seroprevalence study detected bovine enterovirus 1 reactive antibodies in 33% of human blood samples [75]. 

The knowledge about enteric virus infections in commensal rodents and Norway rats in particular is scarce. In this study, we analyzed the occurrence, detection rate, and the diversity of different viral pathogens that are transmitted by the fecal-oral route in Norway rats.

## 2. Materials and Methods

### 2.1. Collection of Norway Rats

Norway rats were previously collected within pest control measures at one site in Hungary (*n* = 17), one site in Czech Republic (*n* = 8), and six sites in Germany (*n* = 26). Collection and dissection of these animals have been described previously [76,77]. These rats were previously investigated for infections with *Leptospira* spp., *Rickettsia* spp., orthopoxvirus, rat polyomavirus, *Anaplasma phagocytophilum*, *Neoehrlichia mikurensis*, *Babesia* spp., and *Bartonella* spp. [7,77]. 

### 2.2. Sample Collection

The dissection and collection of different tissue samples was conducted following standard protocols. Tissue samples were taken from mesentery, duodenum, jejunum, ileum, cecum, and colon. Additionally, samples from content of the intestine were taken, if available (e.g., content of cecum, colon, ileum, and feces).

### 2.3. RNA Extraction from Feces and Tissue Samples

For viral RNA extraction, tissue samples and contents of cecum and colon were homogenized in 500 µL phosphate-buffered saline (PBS) using innuSPEED Lysis tubes P and SpeedMill (Analytik Jena, Jena, Germany). The suspensions were clarified by centrifugation and 140 µL of the supernatants were used for RNA extraction by QIAamp Viral RNA Mini Kit (Qiagen, Hilden, Germany), according to the protocol of the manufacturer. 

### 2.4. Viral RNA Detection Screening Strategy 

In the first line screening, intestinal content and feces samples were tested for RNA of astrovirus, rat astrovirus, norovirus, sapovirus, HAV, HEV, and enteroviruses. In case of a positive result, the other samples were also tested for the corresponding viral RNA. 

### 2.5. Astrovirus RNA Detection and Characterization

Astroviruses were detected using a pan-specific semi-nested RT-PCR as previously described [78]. Further characterization of astroviruses was performed by different rat astrovirus specific semi-nested RT-PCRs. For characterization of novel sequences that cluster with the reference sequence KT946733 [79], a RT-semi-nested PCR with primers rAV104, rAV112a, rAV112b, rAV112c, and rAV111 targeting the ORF1/ORF2 junction (Appendix A) using OneStep RT-PCR Kit (Qiagen) was performed. In brief, 2 µL RNA were used in a final reaction volume of 12.5 µL. For the first round RT-PCR, primers rAV111 and rAV104 were used under the following conditions: 30 min at 50 °C, 15 min at 95 °C, followed by 40 cycles of 30 sec at 94 °C, 30 sec at 54 °C, 2 min at 72 °C, and a final extension for 10 min at 72 °C, while primers rAV112a, rAV112b, rAV112c, and rAV104 were used for the second round PCR using HotStarTaq DNA Polymerase (Qiagen) under the following conditions: 15 min at 95 °C, followed by 30 cycles of 30 sec at 94 °C, 30 sec at 54 °C, 60 sec at 72 °C, and a final extension for 10 min at 72 °C.

Novel rat astrovirus sequences that cluster with reference sequence KT946731 [79] were characterized by a semi-nested typing RT-PCR specific for reference sequences of cluster B using primers rAV108, rAV109, and rAV113 targeting ORF1b and the OneStep RT-PCR Kit (Qiagen). In brief, 2 µL RNA were used in a final reaction volume of 12.5 µL. For the first round RT-PCR, primers rAV108 and rAV109 were used under the following conditions: 30 min at 50 °C, 15 min at 95 °C, followed by 40 cycles of 30 sec at 94 °C, 30 sec at 45 °C, 60 sec at 72 °C, and a final extension for 10 min at 72 °C, while primers rAV108 and rAV113 were used for the second round PCR using HotStarTaq DNA Polymerase (Qiagen) under the following conditions: 15 min at 95 °C, followed by 30 cycles of 30 sec at 94 °C, 30 sec at 49 °C, 30 sec at 72 °C, and a final extension for 5 min at 72 °C.

Human astroviruses were genotyped using a semi-nested RT-PCR with primers AV91, AV91d, AV91e, AV91f, AV91g, AV129, AV129a, AV129b, AV130, and AV131 targeting ORF1b (Appendix A) using OneStep RT-PCR Kit (Qiagen) and 2 µL RNA in a final reaction volume of 12.5 µL. For the first round, primers AV91, AV91d, AV91e, AV91f, AV91g, AV130, and AV131 were used under following conditions: 30 min at 50 °C, 15 min at 95 °C, followed by 30 cycles of 30 sec at 94 °C, 30 sec at 53 °C, 45 sec at 72 °C, and a final extension for 5 min at 72 °C, while primers AV129, AV129a, AC129b, AV130, and AV131 were used for the second round PCR using HotStarTaq DNA Polymerase (Qiagen) under the following conditions: 15 min at 95 °C, followed by 30 cycles of 30 sec at 94 °C, 30 sec at 51 °C, 45 sec at 72 °C, and a final extension for 5 min at 72 °C.

### 2.6. Norovirus RNA Detection, Quantification and Genotyping

Norovirus RNA detection, quantification, and genotyping were done as previously described [30]. Norovirus genotyping was performed by amplification and sequencing of ORF1 (RdRp) and ORF2 (capsid) regions using two primer sets as described previously [80]. 

### 2.7. Sapovirus RNA Detection

Detection of sapoviruses was done as previously described [30].

### 2.8. Rotavirus RNA Detection and Quantification

Detection and quantification of RVA was performed using a quantitative RT-PCR (RT-qPCR) as previously described [81]. The RT-qPCR targets a highly conserved region of the NSP4 segment and detects all described human and many, but not all yet described, animal rotaviruses. RVA genotyping was done as previously described [81].

### 2.9. Enterovirus RNA Detection, Genotyping and Virus Isolation

Human enteroviruses were detected using a previously described nested RT-PCR targeting the viral 5′-non-coding region (NCR) [82]. Sequencing and enterovirus typing of the 5′-NCR region was done using the enterovirus typing tool https://www.rivm.nl/mpf/typingtool/enterovirus/ (accessed on 1 December 2020). Enterovirus strains were genotyped using previously described nested RT-PCR assays targeting the *VP1* gene [83]. Virus isolation was attempted as previously described [83].

### 2.10. Hepatitis A Virus RNA Detection

For HAV detection, a RT-nested PCR using primers HAV5, HAV6, HAV7, and HAV8, targeting the *VP1* gene (Appendix A) was performed by OneStep RT-PCR Kit (Qiagen). In brief, 2.5 µL RNA were used in a final reaction volume of 25 µL. For the first round RT-PCR, primers HAV5 and HAV6 were used under the following conditions: 30 min at 50 °C, 15 min at 95 °C, followed by 30 cycles of 30 sec at 94 °C, 30 sec at 42 °C, 30 sec at 72 °C, and a final extension for 5 min at 72 °C, while primers HAV7 and HAV8 were used for the second round PCR using HotStarTaq DNA Polymerase (Qiagen) under the following conditions: 15 min at 95 °C, followed by 30 cycles of 30 sec at 94 °C, 30 sec at 42 °C, 30 sec at 72 °C, and a final extension for 5 min at 72 °C.

### 2.11. Hepatitis E Virus RNA Detection and Full Genome Amplification

For HEV detection, a nested pangenotypic RT-PCR for human-pathogenic HEV using primers HEV40, HEV41, HEV42, and HEV43, targeting the ORF1 (Appendix A) was performed by OneStep RT-PCR Kit (Qiagen). In brief, 2.5 µL RNA were used in a final reaction volume of 12.5 µL. For the first round RT-PCR, primers HEV40 and HEV41 were used under the following conditions: 30 min at 50 °C, 15 min at 95 °C, followed by 40 cycles of 30 sec at 94 °C, 30 sec at 42 °C, 30 sec at 72 °C, and a final extension for 5 min at 72 °C, while primers HEV42 and HEV43 were used for the second round PCR using HotStarTaq DNA Polymerase (Qiagen) under the following conditions: 15 min at 95 °C, followed by 30 cycles of 30 sec at 94 °C, 30 sec at 42 °C, 30 sec at 72 °C, and a final extension for 5 min at 72 °C. Liver samples of 47 of the 51 rats were previously investigated for ratHEV infection [76].

For HEV full-length genome amplification, extracted RNA was used as a template for one-step RT-PCR by OneStep RT-PCR Kit (Qiagen), as well as for cDNA synthesis using SuperScript IV First-Strand Synthesis System (ThermoFisher, Waltham, MA, USA), followed by nested PCR using HotStarTaq DNA Polymerase (Qiagen). Fragments that overlap across the entire HEV genome were amplified using primers listed in Appendix A. To obtain the 5’-NCR the 5’ Rapid Amplification of cDNA Ends (RACE) system, version 2.0 (Invitrogen) was used; the 3’-NCR was amplified by RT-PCR using dT20AGC and rHEV79 primers (Appendix A).

### 2.12. Sequencing and Phylogenetic Analysis

The amplified DNA fragments were submitted to dideoxy-chain termination sequencing using the corresponding PCR primers (Appendix A) and an automated sequencer, ABI 3730 (Applied Biosystems). For phylogenetic analysis, nucleotide sequences were aligned with the MAFFT algorithm in Geneious prime 11.0.4. Trees were reconstructed in MEGA7.0.26, using the best fit models of substitution pattern with the lowest Bayesian information criterion (BIC) score. The reliability of the branching pattern was tested with bootstrapping (1000 replicates). The nucleotide sequences of the viral pathogens of this study are available under the following GenBank accession numbers, astroviruses (AstV): MW795426-MW7955650 (AstV screening); MW795375-MW795381 (AstV ORF1); MW795395-MW795422 (AstV ORF1/2 junction); norovirus: MW759846-MW759476 (NV-GI ORF1); MW759305-MW759306 (NV-GI ORF2); MW759277-MW759280 (NV-GII ORF1); MW759303 (NV-GII ORF2); HEV: MW795382-MW795394 (HEV screening); MW795566-MW795567 (HEV full-length genome); enterovirus: MW958848-MW928849 (VP1); RVA VP7: MW928847. 

## 3. Results

### 3.1. Virus RNA Detection

Intestines of a total of 51 rats were analyzed for the presence of rotavirus, norovirus, astrovirus, sapovirus, enterovirus, HAV, and HEV RNA. At least RNA of one virus was detected in 49 of the 51 (96%) analyzed animals. The overall detection rate of astrovirus, rotavirus, HEV, norovirus, and enterovirus RNA was 90.2% (*n* = 46), 21.6% (*n* = 11), 9.8% (*n* = 5), 7.8% (*n* = 4), and 2% (*n* = 1), respectively. RNA of HAV and sapoviruses was not detected.

RNA of a single pathogen was detected in 32 out of 51 samples (63%), in 16 animals RNA of two viruses was detected (31%) and in one animal (2%) RNAs of three different viruses were detected. Of the animals tested positive for RNA of only one virus, 29 (90.6%) were positive for astrovirus, and one each for norovirus, HEV, and rotavirus (3.1%). In 10 of the 16 animals tested positive for two pathogens, rotavirus and astrovirus RNAs were detected (62.5%), three animals tested positive for astrovirus and HEV RNAs (18.7%), two (12.5%) for astrovirus and norovirus RNAs, and one animal (6.3%) showed a combination of astrovirus and enterovirus RNAs. In addition, one animal was positive for astrovirus, norovirus, and HEV RNAs (Table 1). 

### 3.2. Astrovirus Detection and Characterization

Astroviruses were initially detected by a pan-specific semi-nested-RT-PCR. In 46 out of the 51 tested animals, astrovirus RNA was detected. Sequence analyses of the screening PCR amplicon using NCBI BlastN https://blast.ncbi.nlm.nih.gov/ (accessed on 1 December 2020) revealed that most of the detected sequences were related to sequences belonging to cluster D, as introduced by To et al., 2017 [79] (GenBank accession No.: KT946733). Thirty-four of the AstV sequences obtained from the screening PCR were related to this cluster D. Five of the rat astrovirus sequences obtained from the screening PCR were related to sequences belonging to cluster B (GenBank accession No.: KT946731). 

In one animal, human astrovirus RNA was detected alongside rat astrovirus RNA. Single sequences from ten animals could not be resolved although samples were tested positive in the RT-PCR, likely due to mixed infections or due to low RNA amounts (KS/11/615; KS/11/620; KS/11/621; KS/14/0001; KS/14/0002; KS/14/0003; KS/14/0004; KS/14/0007; KS/10/2581; KS/15/34).

Interestingly, for two animals inconsistent results of the sequence analysis were obtained. The astrovirus sequences obtained from the duodenum of the animal KS/10/2585 were related to rat astrovirus cluster D [79], whereas the sequences from jejunum, mesentery, and feces samples were identified as human astrovirus. The astrovirus sequences from the content of the cecum from the animal KS/11/616 were more similar to reference sequences specific for rat astrovirus cluster B [79], whereas astrovirus sequences obtained from the colon of the same animal were more related to reference sequences from cluster D [79]. 

To characterize the various rat astroviruses and the human astrovirus in more detail, samples were investigated with different specific PCRs that were designed according to reference sequences, which were related to KT946731 and reference sequences related to KT946733 located in ORF1b and ORF1/ORF2 junction region, respectively. 

Forty-two samples, which were either related to KT946733 (cluster D) in the initial screening PCR or were positive in the screening PCR, but without sequencing information, were tested using a PCR targeting the ORF1/ORF2 junction region. In total, sequences were obtained for 20 animals using this PCR (Figure 1). Almost all sequences belong to the same cluster D, which confirmed the results from the sequence analysis of the screening PCR. Additionally, the analysis of the sequences from the ORF1/ORF2 junction region confirmed the results for animal KS/10/2585, which shows rat astrovirus specific sequences within cluster D for samples from duodenum and ileum and human astrovirus sequences for samples isolated from cecum, colon, jejunum, feces, and content of cecum.

Eight samples, which were initially either more related to KT946731 (cluster B) or which were positive for the screening PCR but without sequencing information, were tested using a PCR specific for sequence KT946731, located in ORF1b. Of the eight tested samples, one sample could be sequenced, and phylogenetic analyses of the sequence confirmed its assignment to cluster B (Figure 2). Phylogenetic analysis confirmed that astrovirus sequences obtained from animal KS/10/2585 clustered both with rat-specific AstV sequences belonging cluster B and human astrovirus sequences, in particular HAstV-8.

In summary, astroviruses were the most frequently detected pathogens in the analyzed animals, with 46 animals testing positive for astrovirus RNA. Sequence information could be obtained from 40 of these strains. In most of the animals (n = 39), rat specific astrovirus RNAs were detected, which belong to two different phylogenetic clusters. Rat astrovirus sequences belonging to cluster D were detected in animals, which were trapped in all five regions of the three countries investigated. Rat astrovirus sequences belonging to cluster B were detected in five animals; these animals were collected in three different regions, two were trapped in Budapest (Hungary), two in Hamburg (Germany), and one in Brno (Czech Republic). 

Interestingly, one animal (KS/10/2585) trapped in Hamburg (Germany) was tested positive for rat astrovirus RNA belonging to cluster B and a human astrovirus belonging to genogroup HAstV-8. Phylogenetic analyses of astrovirus sequences from this animal demonstrated controversial results: Human astrovirus sequences were obtained from jejunum, ileum, cecum, colon, content of cecum, and feces (Figure 1 and Figure 2), but sequences from duodenum and ileum belong to cluster D (Figure 1) and sequences obtained from the jejunum belong to cluster B (Figure 2). These results suggest a mixed infection with a human astrovirus and at least one rat specific astrovirus.

### 3.3. RVA Detection and Characterization

RVA RNA was identified in eleven out of the 51 tested animals (21.6%), mostly detected together with AstV RNA (*n* = 10). RVA RNA was only detected in intestinal content (jejunum, cecum, colon, feces), but not in any of the tested tissue samples. All positive samples showed a very low viral RNA load between 3.7 × 10^2^ genome copies/mL and 1.6 × 10^4^ genome copies/mL, with a median of 2.3 × 10^3^ genome copies/mL. Further genotyping resulted in the determination of G-type for the sample with the highest viral load. Phylogenetic analysis of the sequences obtained from the content of the cecum of this animal revealed that this sequence belongs to the G3 type (Figure 3). The sequences clustered with a reference sequence (KX398364), which was detected in a black rat (*Rattus rattus*) from Northern Italy and shared 91.08% pairwise nucleotide sequence identity [53]. Due to the low virus RNA load, for none of the strains the P type could be identified.

### 3.4. HEV Detection, Characterization and Full-Length Genome Amplification 

Presence of HEV RNA was initially tested by a pan-specific, nested-RT-PCR. HEV RNA was detected in five out of the 51 tested animals (9.8%). Three of the positive tested animals were trapped in Hamburg (Germany) and two animals were collected in Budapest (Hungary). Intestine content was tested positive for HEV RNA in all of the five animals. Additionally, HEV RNA was also detected in different tissues in three of the five positive animals; samples from jejunum, mesentery, ileum, and cecum were positive in two animals, whereas only a single sample from the colon was tested positive. Phylogenetic analysis of a 312 nt fragment generated by the screening PCR, showed that all detected HEV sequences clustered with ratHEV reference sequences, but not with HEV-3 sequences (Figure 4). 

All of the determined sequences clustered in the ratHEV genogroup GI, which was proposed by Mulyanto and colleagues in 2014 [84]. Between the HEV sequences derived from different animals, pairwise nucleotide sequence identities of 85.12–90.84% were calculated. 

Full-length genome sequencing was done by amplification of overlapping fragments covering the entire genome. Fragments of the 5′- and 3′-untranslated regions were generated for two animals from Hungary (KS/11/616 and KS/11/620). The length of the complete genome without the poly-A tail of both strains is 6941 nt with a pairwise nucleotide sequence identity of the entire genome of 85.89% (Figure 5). The three major open reading frames ORF1, ORF2, and ORF3 could be identified. The putative ORF4 was also predicted in both full-length genomes at nt positions 27–578 overlapping the ORF1 in a +1 frame. The lengths of the ORF1-, ORF2-, ORF3-, and ORF4-encoded proteins of both strains detected in this study were identical with 1634, 644, 103, and 183 amino acid residues, respectively. The strain KS/11/616 showed 88.4% pairwise nt sequence identity about the entire genome with a ratHEV (JN167538) strain detected in Germany [61], whereas strain KS/11/620 is more related to another ratHEV strain (GU345042) from Germany with a pairwise nt sequence identity of 88.8% (Table 2) [16]. The highest pairwise nt sequence identity was detected in ORF4, whereas the highest identity on amino acid level was detected in the ORF2-encoded capsid protein. The lowest pairwise nt and amino acid sequence identities for both ratHEV strains from this study was detected in ORF1 and ORF3-encoded protein, respectively. 

### 3.5. Norovirus Detection and Characterization

Noroviruses were detected using a RT-real-time PCR, which differentiates between genogroup GI and genogroup GII. All positive and borderline positive samples were further genotyped using nested RT-PCRs targeting the ORF1 and the ORF2 of the norovirus genome. Of the 51 analyzed animals, norovirus RNA was detected in four animals (7.8%). In one animal, a norovirus belonging to genogroup GI was detected, whereas the other three animals were positive tested for noroviruses that belong to genogroup GII. Noroviruses could be detected in the content of cecum in all four animals and additionally in ileum, cecum, and feces in one animal. Phylogenetic analysis revealed norovirus genotypes GI.P4-GI.4 and GII.P33-GII.1 in two animals. In another animal, only the ORF1 sequence could be determined to belong to GII.P21 (Figure 6 and Figure 7). Genotyping failed in one animal infected with genogroup GII norovirus, due to negative genotyping PCRs.

### 3.6. Enterovirus Detection and Characterization

Human enterovirus RNA was detected in one of the 51 tested animals (KS/10/2572) trapped in Hamburg (Germany). Enterovirus RNA was detected in cecum and the content of the cecum by using a nested RT-PCR targeting the 5′-NCR of the virus. Sequencing analysis using the enterovirus typing tool revealed that in the cecum a species *Enterovirus A* strain was detected, whereas in the cecum content an *Enterovirus C* strain was identified. An enterovirus could be isolated using RD-A cells from a sample obtained from the content of the cecum. Species specific nested RT-PCRs were performed using the samples obtained from the cecum tissue, from the content of the cecum and from the virus isolate. VP1 specific amplicons could be generated from the virus isolate and from the sample obtained from the content of cecum. Sequencing and phylogenetic analysis allowed typing of this strain and revealed presence of coxsackievirus A20 (CVA20) (Figure 8). The nucleotide sequences from enterovirus isolate and from the original material were almost identical (pairwise sequence identity of 99.7%). Both sequences differed from each other at 3 nucleotides. The typing results of the screening PCR in the 5′-NCR may be a hint that this animal was infected with two different enterovirus strains.

## 4. Discussion

The knowledge about the presence and diversity of enteric viral pathogens in rodents is so far limited. The present study reports about the detection rate and genetic diversity of various viral pathogens in Norway rats from three European countries (Germany, Hungary, and Czech Republic). The intestines of 51 animals were analyzed for the presence of astro-, noro-, sapo-, rota-, enteroviruses, HEV, and HAV using virus-specific RT-qPCR (noro-, rota-, and sapovirus) or virus-specific nested RT-PCR (astro-, enterovirus, HEV, and HAV). Viral RNA positive samples were further characterized by sequence determination and phylogenetic analysis. 

Our data show a high detection rate of these viral pathogens, with at least RNA of one virus detected in 49 of 51 animals analyzed, RNAs of two viral pathogens were simultaneously detected in 16 animals, and RNAs of three viral pathogens were detected in one animal. Mixed infections with various viral pathogens have been previously detected also in fecal samples of Norway rats from Berlin and New York City [18,19]. 

Astrovirus infections of rats have been reported in the past. NGS-based studies revealed astrovirus sequences in Norway rats from Berlin and New York City [18,19]. In a study from China, 562 fecal samples from rodents were analyzed and astroviruses were detected in 11.9% of the animals [79]. In another study, 625 small mammals from 11 different species, including 19 Norway rat specimens, were tested for astroviruses in Singapore. Astroviruses were found in 14% of the small mammals, with the highest detection rate up to 26.7% in *Rattus* spp. [29]. Indeed, astroviruses were the most frequently detected viruses in the present study, with 46 of the 51 animals tested positive for astrovirus RNA (90.2%). Most of the detected astrovirus sequences could be assigned to two distinct clusters together with reference sequences from rat astroviruses. In addition to the rat-specific astroviruses, human pathogenic astrovirus of genotype HAstV-8 was detected in one animal. To our knowledge, this is the first detection of a human astrovirus in a Norway rat. Despite the broad distribution of astroviruses in different species, zoonotic infections are rarely described. In a study in which 879 fecal samples from free-living and zoo-dwelling non-human primates were examined for astroviruses, 7.7% of the animals tested positive. Of the detected astrovirus sequences, 60% could be assigned to the human pathogenic genotype HAstV-8 [85]. Additionally, an infection with a canine astrovirus has been detected in a child with acute gastroenteritis [30].

In the presented study, rotavirus RNA was detected in 22% of the animals (*n* = 11). This is in line with a previously published study where RVA RNA was detected in four out of 20 (20%) Norway rat fecal samples, collected in Berlin (Germany), using also a RT-qPCR detection assay [18]. In our study, genotyping could only be successfully performed on one sample due to the overall low viral RNA load between 3.7 × 10^2^ and 1.6 × 10^4^ genome copies/mL. The phylogenetic analysis of this sequence shows that this strain is related to rat rotavirus and belongs to the genotype G3; a determination of the P type was not possible in this sample, due to the low viral RNA load. The phylogenetic analysis indicates that the sequence is a rat specific RVA sequence, as the VP7 sequence has a nucleotide sequence identity of 91.1% to a reference sequence obtained from a rat [53]. This study in Italy detected a RVA strain in a black rat that has an unusual genome constellation: G3-P [3]-I1-R11-C11-M10-A22-N18-T14-E18-H13. This strain could be an interspecies reassortant, as the RVA genome segments VP1-4, VP7, NSP1, NSP3, NSP4, and NSP5 are closely related to those of rodent RVA, while NSP2 and VP6 are closely related to human RNA [53]. 

In our study, we detected HEV RNA in five animals (10%), all obtained sequences were identified as ratHEV. The absence of HEV-3 sequences in the investigated Norway rat samples is in line with previous studies on Norway rat liver samples from Germany and other European countries [16,76]. Human pathogenic HEV-3 genotype has been only rarely detected in Norway rats [86,87]; recently in one Norway rat from Belgium, a short rabbit HEV-like genotype 3 sequence was detected [76]. The detection of ratHEV has been described from a variety of countries in Europe, Asia, and Northern America [16,61,76,84,88,89,90,91]. The detection rates of rat HEV in *Rattus* spp. in different studies is highly variable. In a study from China, ratHEV RNA was detected in 58% of the Norway rats and 47% in *Rattus tanezumi* tested [90]. In different studies from Europe and Asia, ratHEV RNA was detected in 10 to 14% of the tested rats [61,76,88,91]. Thus, the 10% detection rate found in this study is in the same range. 

In this study, we detected ratHEV RNA in animals collected in Hungary and Germany. In a previous study, liver samples of 47 of the 51 animals investigated in this study were analyzed for the presence of HEV RNA [76]. The results of the previous study could be confirmed here for 38 of the analyzed animals, but discrepant results were detected in nine of the investigated animals. In two of these animals, ratHEV RNA was detected in the intestine, whereas no RNA has been detected in the liver of these animals before. On the other hand, ratHEV RNA could be detected in seven of these nine animals in the liver, while no ratHEV RNA was detectable in the intestine of these animals. The results of these two studies demonstrate the importance of the selection of organs for the detection of ratHEV RNA. However, the discrepancies could also be explained by differences in the sensitivity of the detection methods used.

Phylogenetic analysis revealed that all of the obtained ratHEV sequences in this study belong to the genotype GI, which was first proposed by Mulyanto and colleagues in 2014 [84]. Full-length genome analysis from two animals in our study revealed that both ratHEV strains showed a quite low pairwise nt sequence identity although both animals were collected in Budapest within the same time frame (May to June 2010). This confirms that different ratHEV strains simultaneously circulate in rat populations, as previously reported [76]. Both novel ratHEV strains from Budapest showed the typical genome organization with the three major ORFs (ORF1, ORF2, ORF3) and the putative ORF4, which was first described by Johne et al. [61]. The predicted size of the ORF1-encoded non-structural protein precursor is 1634 amino acid residues and thereby similar to values reported for ratHEV strains detected in Indonesia and Germany [61,84]. In contrast, other ratHEV strains from Germany, Indonesia, and Vietnam were characterized by quite different ORF1-encoded precursor protein lengths [16,54,84]. The lengths of the other viral proteins were identical between strains detected in this study and the ratHEV reference strains. 

In our study, human norovirus RNA was detected in four of the 51 animals tested (8%). This detection rate is in a similar range like that reported from a study on fecal samples from rodents in Finland [37]. While only 2% of the rodent samples tested positive, human noroviruses were found in 27% of the bird-derived samples. In a NGS-based study in two of 20 Norway rats from Berlin, norovirus reads were identified, whereas in another NGS based study with fecal samples from 133 Norway rats collected in New York City no norovirus reads were reported [18,19]. Three of the four norovirus RNA positive samples from our study belonged to genogroup II, which is responsible for the majority of norovirus infections in humans. One sample belongs to genogroup I, which is also human pathogenic but less frequently detected in humans compared with genogroup II [92]. Three of the four norovirus RNA positive samples could be further genotyped. In two samples, the genotype could be determined from ORF1 and ORF2 sequences, with one strain belonging to genotype GI.P4-GI.4, while the other strain being a recombinant variant GII.P33-GII.1. In the third sample, only the genotype determined from ORF1 sequence could be identified as GII.P21. Knowledge of the diversity of zoonotic norovirus infections is very limited because genotyping of detected viruses is often difficult due to low viral loads. However, there are some reports about the genetic diversity of zoonotic norovirus infections. In a Norway rat, trapped in Copenhagen, Denmark, RNA of a recombinant norovirus GI.P11-GI.6 was detected [15]. In one dog, infection with the recombinant strain GII.P31-GII.4 Sydney was detected [40], viruses of genotype GII.17 were described in rhesus monkeys [43], and viruses of genotypes GII.3 and GII.4 were described in birds [37]. 

In our study, enterovirus RNA was detected in one of the analyzed animals. In a previously published NGS-based study, enterovirus reads were detected in four out of 20 Norway rats, collected in Berlin (Germany), with one animal showing a higher number of enterovirus reads [18]. Phylogenetic analysis of the 5’-NCR showed that an enterovirus A strain was identified in the cecum, whereas an enterovirus C strain was detected in the contents of cecum. Infectious enterovirus particles could be isolated using RD5 cells in the sample obtained from the content of cecum, indicating an active replication of the enterovirus in the intestine of Norway rat. Genotyping of the VP1 region was possible from the isolated enterovirus, which was identified as a coxsackievirus A20 strain. The results of the phylogenetic analysis of the 5’-NCR suggest a co-infection of two different enteroviruses in this animal. To date, there have been few reports of zoonotic infections with enteroviruses. For example, human enterovirus infection has been detected in nonhuman primates [71,72,73]. To date, detection of human enteroviruses in rodents has not been described. 

## 5. Conclusions

Our study provides novel insights into the diversity and detection rate of different viral pathogens in Norway rats in three different countries in Europe. Our results showed a high detection rate of 96% in the tested animals for RNAs of five different enteric viral pathogens, all with high zoonotic potential to human transmission. Astrovirus RNA was most commonly detected in this study. Phylogenetic analysis revealed RNAs of human pathogenic viruses in five of the tested animals, with norovirus RNA in four, astrovirus RNA in one, and enterovirus infection also in one animal. This study demonstrates RNA of zoonotic human enteroviruses in rodents for the first time. We could show the presence of infectious enterovirus particles, suggesting a replicative infection of a Norway rat with a coxsackievirus A20 strain. The detection of HEV, astrovirus, norovirus, and enterovirus RNAs in the various intestinal tissues isolated from these animals may suggest a natural path of infection and indicate replication of the viruses in the respective tissues. However, we cannot exclude the possibility that the detection of some of the viral pathogens, with the exception of enterovirus, from the contents of the intestine are due to animals having ingested corresponding viruses through contaminated food. The presence of infectious virus particles in these tissues would need to be investigated in future projects. 

## Figures and Tables

**Figure 1 viruses-13-00992-f001:**
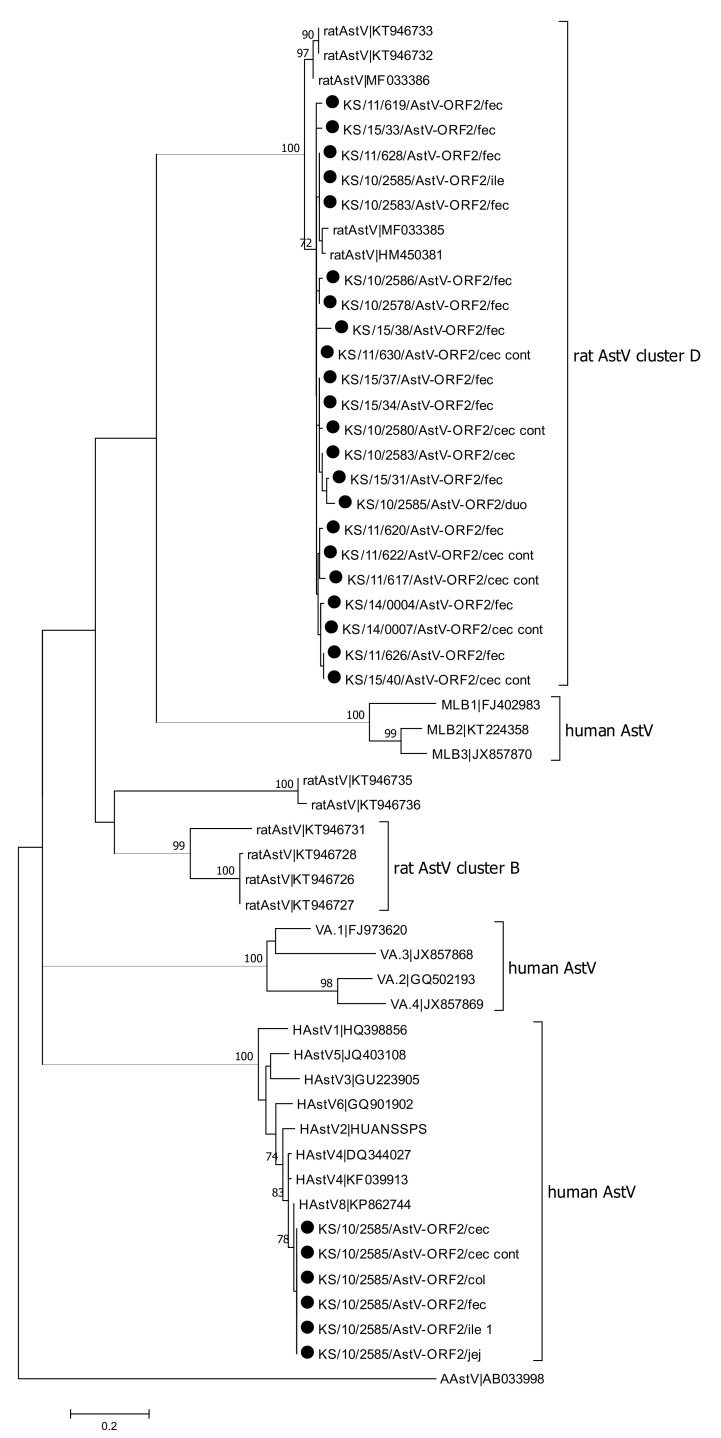
Phylogenetic tree of a 970 nucleotide (nt) alignment of the ORF1/ORF2 junction region of the novel astrovirus (AstV) strains detected in animals of this study (denoted by dot) and AstV reference sequences (accession numbers are indicated). The tree was reconstructed using the Maximum Likelihood method based on the Kimura 2-parameter model with Bootstrap test (1000 replicates). Bootstrap values above 70 are shown. The bar indicates the number of substitutions per site. An avian astrovirus sequence (AAstV|AB033998) was used as outgroup. Abbreviations: cec = cecum; cont = content; col = colon; duo = duodenum; fec = feces; ile = ileum; jej = jejunum.

**Figure 2 viruses-13-00992-f002:**
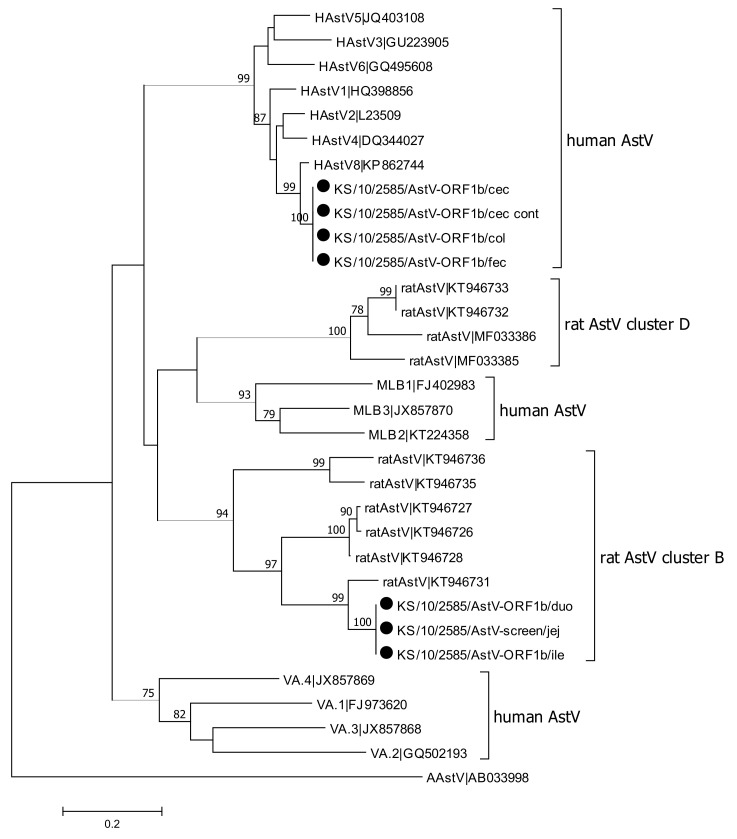
Phylogenetic tree of a 581 nt alignment of ORF1b region of the novel astrovirus (AstV) strains detected in this study (denoted by dot) and AstV reference sequences (accession numbers are indicated). The tree was reconstructed using the Maximum Likelihood method based on the Tamura-3-parameter model with Bootstrap test (1000 replicates). Bootstrap values above 70 are shown. An avian astrovirus sequence (AAstV|AB033998) was used as outgroup. The bar indicates the number of substitutions per site. Abbreviations: cec = cecum; cont = content; col = colon; duo = duodenum; fec = feces; ile = ileum; jej = jejunum.

**Figure 3 viruses-13-00992-f003:**
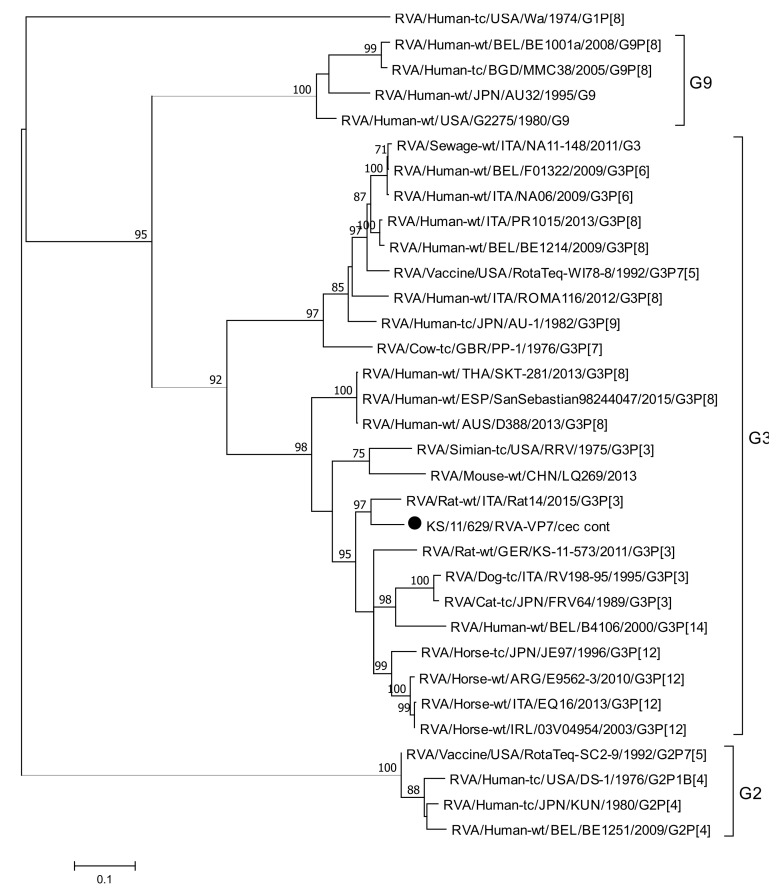
Phylogenetic analysis of a 820 nt fragment of *VP7* gene from the novel rotavirus strain identified in this study (marked with a dot) and reference sequences. The phylogenetic tree was reconstructed using the Maximum Likelihood method and Tamura–Nei parameter with Bootstrap test (1000 replicates) method available in MEGA7. The bar indicates the number of substitutions per site. Bootstraps values above 70 are shown. Abbreviations: cec cont = content of cecum.

**Figure 4 viruses-13-00992-f004:**
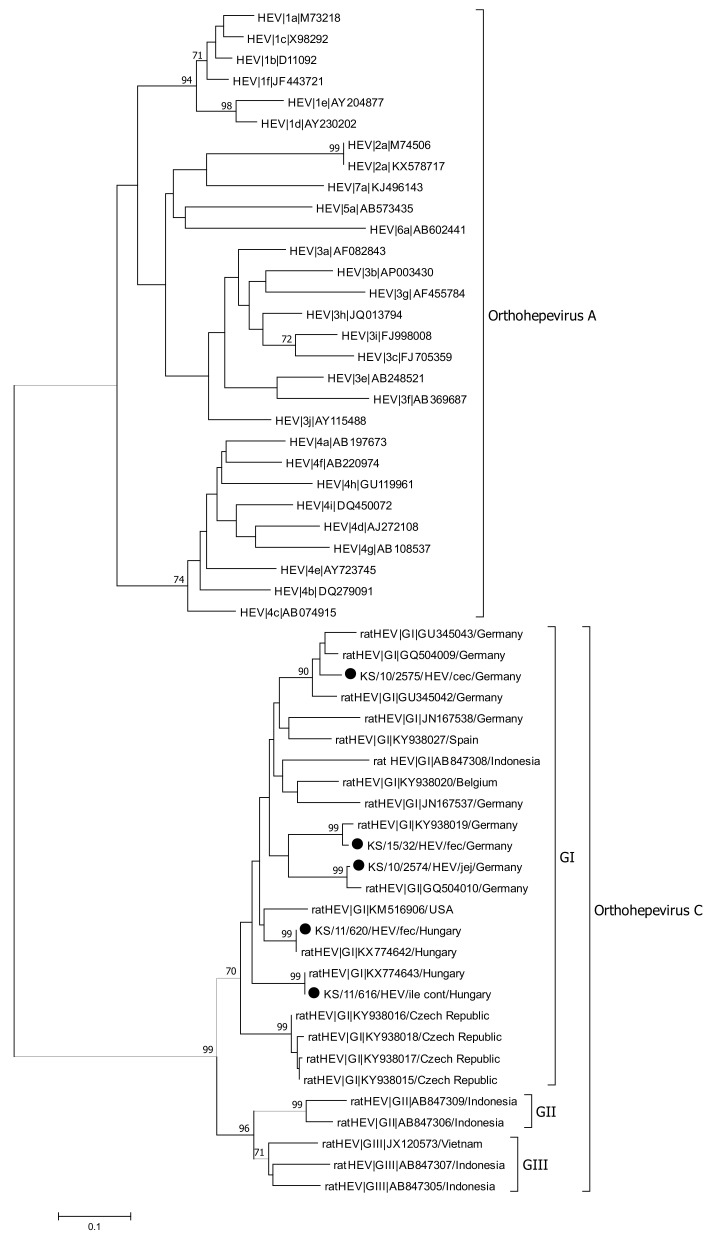
Phylogenetic analysis of a 312 nt fragment of ORF1 from HEV strains detected in this study (marked with dot) and reference sequences of species *Orthohepevirus C* and *Orthohepevirus A*. The phylogenetic tree was reconstructed using the Neighbor-Joining method with Bootstrap test (1000 replicates) and the Tamura–Nei parameter method available in MEGA7. The bar indicates the number of substitutions per site. Bootstraps values above 70 are shown. Abbreviations: cec = cecum; cont = content; fec = feces; ile = ileum; jej = jejunum.

**Figure 5 viruses-13-00992-f005:**
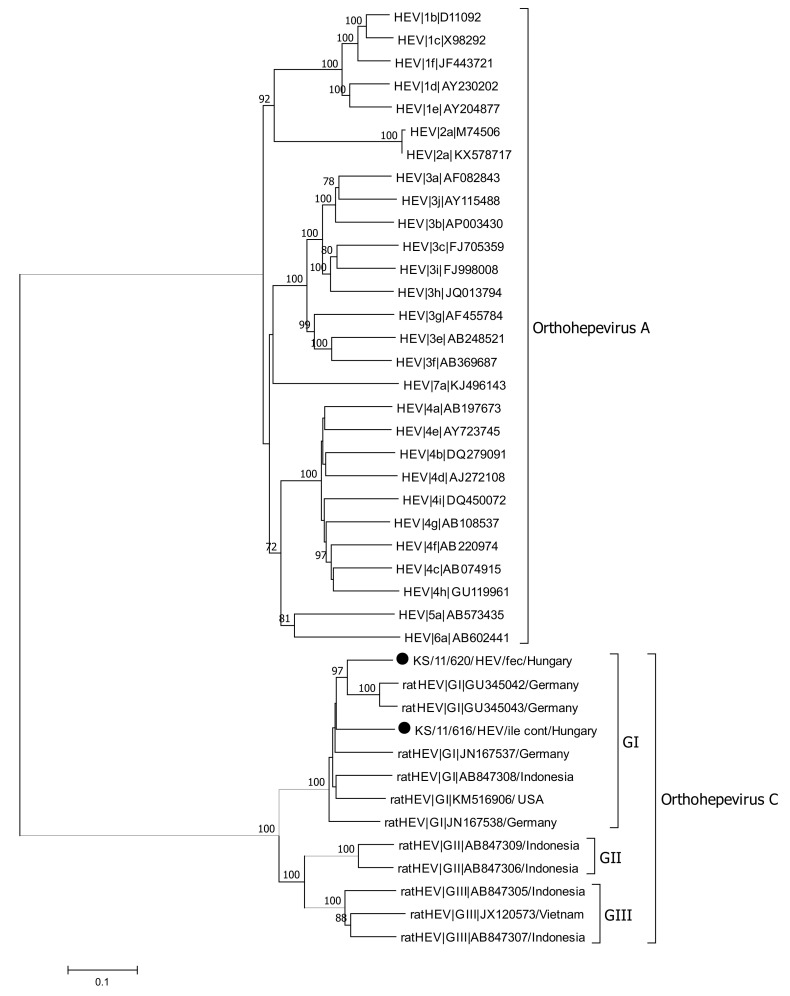
Phylogenetic analysis of the entire genome from ratHEV strains detected in this study (marked with dot) and reference sequences of representative strains of species *Orthohepevirus C* and *Orthohepevirus A*. The phylogenetic tree was reconstructed using the Neighbor-Joining method with Bootstrap test (1000 replicates) and the Tamura–Nei parameter method available in MEGA7. The bar indicates the number of substitutions per site. Bootstraps values above 70 are shown. Abbreviations: cont = content; fec = feces; ile = ileum.

**Figure 6 viruses-13-00992-f006:**
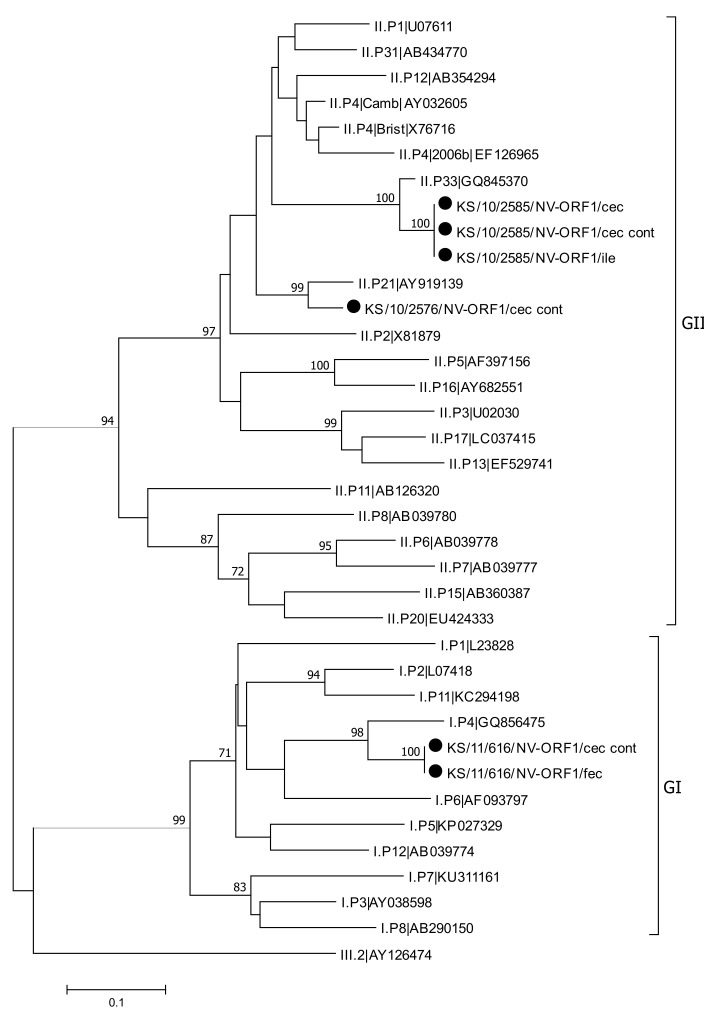
Phylogenetic analysis of a 263 nt fragment of ORF1 from norovirus strains detected in this study (marked with dot), human norovirus reference sequences and a single sequence that was detected in a rat from Copenhagen, Denmark (GI.P11|KC294198). The phylogenetic tree was reconstructed using the Neighbor-Joining method with Bootstrap test (1000 replicates) and the Kimura 2-parameter method available in MEGA7. The bar indicates the number of substitutions per site. Bootstraps values above 70 are shown. The bovine norovirus sequence III.2|AY126474 was used as an outgroup. Abbreviations: cec = cecum; cont = content; fec = feces; ile = ileum.

**Figure 7 viruses-13-00992-f007:**
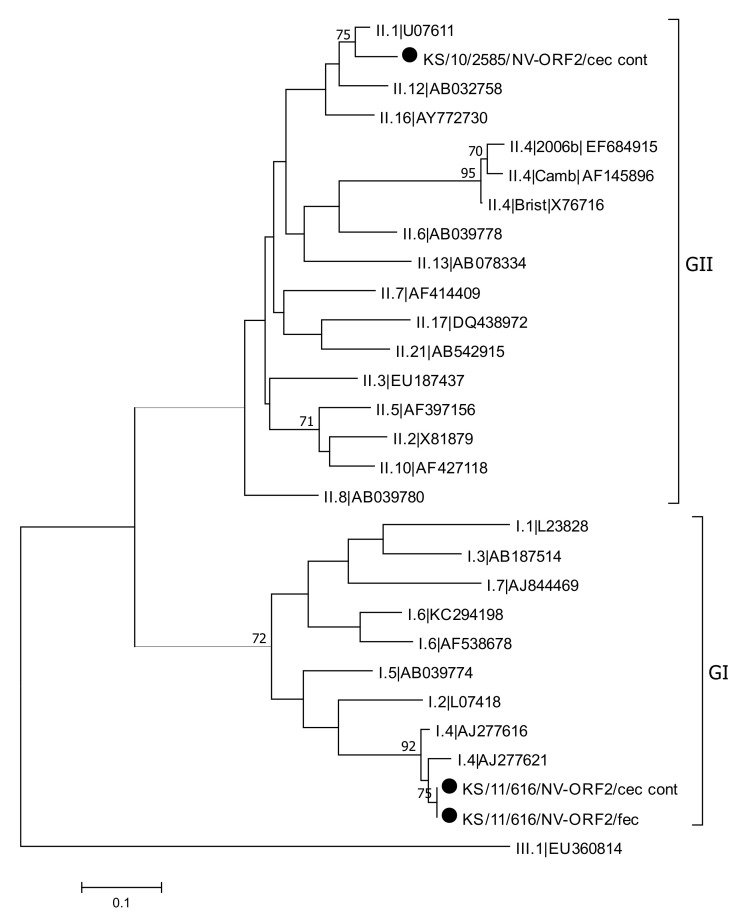
Phylogenetic analysis of a 255 nt fragment of ORF2 from norovirus strains detected in this study (marked with a dot), human norovirus reference sequences and a single sequence that was detected in a rat from Copenhagen, Denmark (GI.6|KC294198). The phylogenetic tree was reconstructed using the Neighbor-Joining method with Bootstrap test (1000 replicates) and the Kimura 2-parameter method available in MEGA7. The bar indicates the number of substitutions per site. A bovine norovirus sequence (GIII.1|EU360814) was used as an outgroup. Bootstraps values above 70 are shown. Abbreviations: cec = cecum; cont = content; fec = feces.

**Figure 8 viruses-13-00992-f008:**
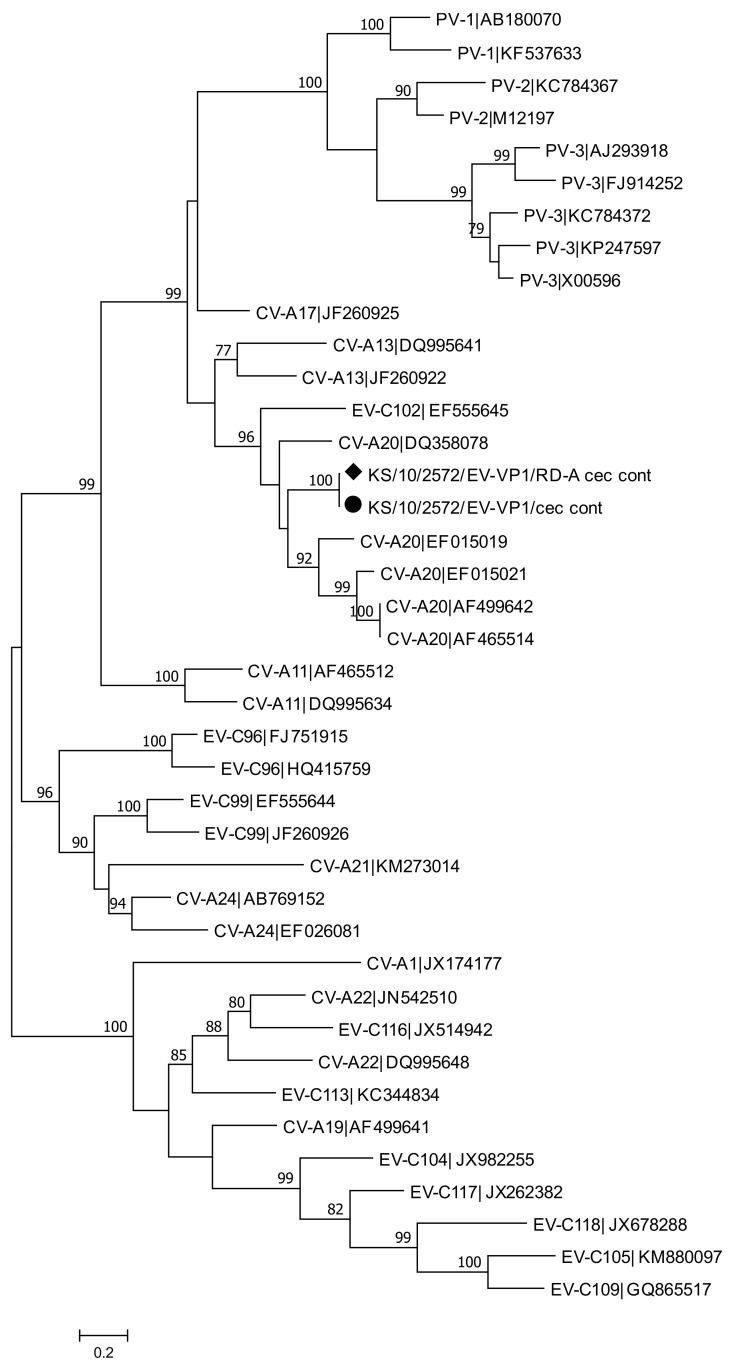
Phylogenetic analysis of a 737 nt fragment of *VP1* gene from the enterovirus strain (marked with a dot) and the enterovirus isolate from this study (marked with a rhombus) and human enterovirus reference sequences. The phylogenetic tree was reconstructed using Maximum Likelihood method with General-Time-Reversible model with Bootstrap test (1000 replicates) available in MEGA7. The bar indicates the number of substitutions per site. Bootstraps values above 70 are shown. Abbreviations: cec = cecum; cont = content.

**Table 1 viruses-13-00992-t001:** Viral RNA detection in Norway rats of this study.

Animal ID	Country	City/Region	AstV	RVA	HEV	NV	EV
KS/11/613	Hungary	Budapest (CH)				x	
KS/11/614	Hungary	Budapest (CH)	x				
KS/11/615	Hungary	Budapest (CH)	x				
KS/11/616	Hungary	Budapest (CH)	x		x	x	
KS/11/617	Hungary	Budapest (CH)	x				
KS/11/618	Hungary	Budapest (CH)	x	x			
KS/11/619	Hungary	Budapest (CH)	x	x			
KS/11/620	Hungary	Budapest (CH)	x		x		
KS/11/621	Hungary	Budapest (CH)	x				
KS/11/622	Hungary	Budapest (CH)	x	x			
KS/11/623	Hungary	Budapest (CH)	x				
KS/11/624	Hungary	Budapest (CH)	x				
KS/11/626	Hungary	Budapest (CH)	x				
KS/11/627	Hungary	Budapest (CH)					
KS/11/628	Hungary	Budapest (CH)	x				
KS/11/629	Hungary	Budapest (CH)	x	x			
KS/11/630	Hungary	Budapest (CH)	x				
KS/14/0001	Czech Republic	Brno (SM)	x	x			
KS/14/0002	Czech Republic	Brno (SM)	x	x			
KS/14/0003	Czech Republic	Brno (SM)	x				
KS/14/0004	Czech Republic	Brno (SM)	x				
KS/14/0005	Czech Republic	Brno (SM)	x	x			
KS/14/0006	Czech Republic	Brno (SM)	x				
KS/14/0007	Czech Republic	Brno (SM)	x	x			
KS/14/0008	Czech Republic	Brno (SM)	x				
KS/10/2572	Germany	Hamburg	x				x
KS/10/2573	Germany	Hamburg	x				
KS/10/2574	Germany	Hamburg	x		x		
KS/10/2575	Germany	Hamburg	x		x		
KS/10/2576	Germany	Hamburg	x			x	
KS/10/2577	Germany	Hamburg	x				
KS/10/2578	Germany	Hamburg	x				
KS/10/2579	Germany	Hamburg	x				
KS/10/2580	Germany	Hamburg	x				
KS/10/2581	Germany	Hamburg	x				
KS/10/2582	Germany	Hamburg					
KS/10/2583	Germany	Hamburg	x				
KS/10/2584	Germany	Hamburg	x				
KS/10/2585	Germany	Hamburg	x			x	
KS/10/2586	Germany	Hamburg	x				
KS/10/2587	Germany	Hamburg	x				
KS/15/31	Germany	Hamburg	x	x			
KS/15/32	Germany	Hamburg			x		
KS/15/33	Germany	Hamburg	x				
KS/15/34	Germany	Hamburg	x	x			
KS/15/35	Germany	Ahlen (NRW)		x			
KS/15/36	Germany	Dortmund (NRW)	x				
KS/15/37	Germany	Stahlbrode (MWP)	x				
KS/15/38	Germany	Dörenhagen (NRW)	x				
KS/15/39	Germany	Dörenhagen (NRW)	x				
KS/15/40	Germany	Elmenhorst (MWP)	x				

AstV, astrovirus; RVA, rotavirus A; HEV, hepatitis E virus; NV, norovirus; EV, enterovirus; CH, Central Hungary; SM, South Moravia; NRW, North Rhine-Westphalia; MWP, Mecklenburg-Western Pomerania.

**Table 2 viruses-13-00992-t002:** Nucleotide and deduced amino acid sequence identity between ratHEV full-length genomes from this study, compared with different ratHEV reference strains detected in Germany, Indonesia, and Vietnam.

			Pairwise Identity nt (%)					Pairwise Identity aa (%)			
	Strain ID/Accession No.	GT	Entire Genome	ORF 1	ORF 2	ORF 3	ORF 4	ORF 1	ORF 2	ORF 3	ORF 4
KS/11/616											
	JN167538	GI	88.4	87.6	90.5	95.1	96.0	96.5	98.3	91.2	90.1
	GU345042	GI	86.8	85.8	89.7	93.5	94.4	94.8	97.5	85.3	85.3
	AB847306	GII	77.2	75.8	80.3	80.9	91.7	87.6	98.3	65.7	79.2
	JX120573	GIII	76.5	75.2	79.4	79.3	90.4	87.4	92.0	63.7	76.5
KS/11/620											
	JN167538	GI	86.7	85.5	89.8	93.2	94.9	94.3	97.4	89.2	86.3
	GU345042	GI	88.8	87.7	91.6	93.2	94.2	94.6	96.5	89.2	86.3
	AB847306	GII	77.0	75.8	79.7	78.0	92.4	87.1	91.6	61.8	80.3
	JX120573	GIII	76.6	75.1	79.6	81.2	91.3	87.5	92.1	66.7	79.2

Abbreviations: GT = genotype; nt = nucleotide; aa = amino acid.

## Data Availability

All data are given in the manuscript and the Appendix A. All novel nucleotide sequences are uploaded at GenBank.

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
