# Peer review of "Presence and Diversity of Different Enteric Viruses in Wild Norway Rats (Rattus norvegicus)"

_viruses, 2021, doi:10.3390/v13060992_

Round 1

Reviewer 1 Report

Rodents are important reservoirs for many different pathogens and involved in the emergence and dissemination of human pathogenic viruses. The knowledge about the presence and diversity of enteric viral pathogens in rat is so far limited. This study reports about the detection rate and genetic diversity of various viral pathogens in Norway rats from three European countries. The results are of interest and the experiments are well performed. However, some issues need to be clarified before publication:

Major revision

  1. line 216-220, It is suggested that three different primer sets should be specified from reference 80.
  2. line 216-220, Primer sets of reference 80 are specific for human RVA. Are those primer sets also pan-specific for animal RVA?
  3. line 277-282, GenBank accession numbers are now non-available on internet.
  4. Line 410-412, RVA RNA genome copies in intestinal content have been measured. Can you also show genome copies for other viruses?
  5. Line 219, Norovirus genotyping was performed by amplification and sequencing in region A (RdRp) and region C (junction of ORF1 and ORF2) using three different primer sets. However, further genotyping uses nested RT-PCRs targeting the ORF1 and the ORF2 of the norovirus genome (Line 497-498). Which target sites are used for genotyping of norovirus.

6. Because, you cannot exclude the possibility that animals having ingested corresponding viruses through contaminated food in conclusions. It is suggested that you should carefully use the term of “infect” to describe viral RNA detection of human viruses in rat samples except coxsackievirus A-20 isolate.

Author Response

Dear Reviewer,

we are very grateful for your very heplful comments. Here is a point-by-point response to the comments and a summary of the changes made to the manuscript.

  1. line 216-220, It is suggested that three different primer sets should be specified from reference 80.

We are very grateful for this helpful comment. In this study two different sets of primers were used, located in the ORF1 (region A) and ORF2 (region C). Both PCR were described in the reference 80. In this reference is a third set of primer described which targets in ORF2 the P2 domain. But this PCR was not used in the presented study. We have clarified this part of the manuscript (line 217-220)

  1. line 216-220, Primer sets of reference 80 are specific for human RVA. Are those primer sets also pan-specific for animal RVA?

The RVA RT-qPCR is described in reference 81. The primer pair and the probe are located in a highly conserved region of the NSP4 segment. The PCR is validated for human RVA sequences and detects all known human RVA strains and many, but not all animal RVA strains. We have re-written this part of the manuscript (line 226-229). 

  1. line 277-282, GenBank accession numbers are now non-available on internet.

Yes, you are right, the sequences will be released after publication of the manuscript by the GenBank.

  1. Line 410-412, RVA RNA genome copies in intestinal content have been measured. Can you also show genome copies for other viruses?

This is a very helpful comment. The detection of norovirus RNA was also done by using a RT-qPCR. But this PCR is optimized and validated for human samples which are normally characterized by high viral load. In contrast, the analyzed samples in the presented study are characterized by a low virus load. Unfortunately, in this high cp-values the standard curve, which is used for the quantification is not in a linear rage. Therefore, we can not present the viral RNA load for norovirus. The other pathogens (astrovirus, HEV and enterovirus) were detected by using a RT-nested-PCR.

  1. Line 219, Norovirus genotyping was performed by amplification and sequencing in region A (RdRp) and region C (junction of ORF1 and ORF2) using three different primer sets. However, further genotyping uses nested RT-PCRs targeting the ORF1 and the ORF2 of the norovirus genome (Line 497-498). Which target sites are used for genotyping of norovirus.

Thank you for this comment. We used in the presented study two different set of primers to detect also recombinant norovirus strain. The first PCR is located in ORF1 (region A) and the second is located in ORF2 (region C). To make it easier for the readership we harmonized this result section with the material and method section, by writing only ORF1 and ORF2. We hope that this is easier to understand.

  1. Because, you cannot exclude the possibility that animals having ingested corresponding viruses through contaminated food in conclusions. It is suggested that you should carefully use the term of “infect” to describe viral RNA detection of human viruses in rat samples except coxsackievirus A-20 isolate.

We are very grateful for this very helpful comment. We had point this out in the limitation of the manuscript. In the revised version of the manuscript we have re-written the corresponding parts throughout the hole manuscript. We hope that this clarifies that we detected the RNA of the virus and that this not necessarily means that the animal is infected with the virus, with the exception of the shown real infection with the coxsackievirus-A20.

Reviewer 2 Report

I would like to congratulate the authors for the immense amount of work presented in a with a well-written paper. 

It is a long read, though, but there is no simple way to present such detailed and extensive work.

page 1 lane 38 misses the word 'are' (and are involved) and on page 9 in the legend to Figure 1 I spotted that the word parameter is broken up wrongly. 

One comment that I would have and that the authors refer to is that finding a sequence of a certain virus does not necessarily mean that it is infectious. It is certainly a point that needs more attention when trying to understand the impact of rodents and other wild or domestic animals in zoonotic transmission of viruses. 

Author Response

Dear Reviewer,

we are very thankful for your very helpful comments. Here is a point-by-point response to the comments and a summary of the changes made to the manuscript.

I would like to congratulate the authors for the immense amount of work presented in a with a well-written paper. 

It is a long read, though, but there is no simple way to present such detailed and extensive work.

Thank very much for this very appreciative comment.

page 1 lane 38 misses the word 'are' (and are involved) and on page 9 in the legend to Figure 1 I spotted that the word parameter is broken up wrongly. 

Thank you for this helpful comment we have changed both accordingly.

One comment that I would have and that the authors refer to is that finding a sequence of a certain virus does not necessarily mean that it is infectious. It is certainly a point that needs more attention when trying to understand the impact of rodents and other wild or domestic animals in zoonotic transmission of viruses. 

We are very grateful for this very helpful comment. We had point this out in the limitation of the manuscript. In the revised version of the manuscript we have re-written the corresponding parts throughout the hole manuscript. We hope that this clarifies that we detected the RNA of the virus and that this not necessarily means that the animal is infected with the virus, with the exception of the shown real infection with the coxsackievirus-A20.

Round 2

Reviewer 1 Report

No comment and suggestion for authors